# Experiential Learning for Sustainability in Supply Chain Management Education

David Ernesto Salinas-Navarro [1,*], Christopher Mejia-Argueta [2], Luis Montesinos [3,4,*] and Ericka Z. Rodriguez-Calvo [5]

1  Aston Business School, Aston University, Birmingham B4 7ER, UK
2  Center for Transportation & Logistics, Massachusetts Institute of Technology, Cambridge, MA 02142, USA
3  Institute of Advanced Materials for Sustainable Manufacturing, Tecnologico de Monterrey, Mexico City 14380, Mexico
4  School of Engineering and Sciences, Tecnologico de Monterrey, Mexico City 14380, Mexico
5  School of Engineering and Sciences, Tecnologico de Monterrey, Puebla City 72810, Mexico
*  Correspondence: d.salinas-navarro@aston.ac.uk (D.E.S.-N.); lmontesinos@tec.mx (L.M.)

**Abstract:** This work is about sustainability-related learning experiences for the discipline of supply chain management (SCM) in Higher Education. It arises from the need to motivate students with relevant and interesting activities to improve their learning performance. Higher Education must respond to dynamic demands to keep impactful topics for students, organizations, and society over time. This work addresses the relevance of contemporary challenges in real-world SCM situations concerning Sustainable Development Goals (SDGs). It also provides an actionable framework integrating experiential learning ideas, the ADDIE model for instructional design, the Triple Bottom Line for sustainability, the continuous improvement cycle, and the SDGs into an SCM model. In a case study, the article illustrates the use of this framework for instructional design in a learning experience from an undergraduate course in an Industrial and Systems Engineering program. The application describes the impact of food ecosystems on cities and communities during the COVID-19 crisis. The results suggest positive attainment levels in students' learning outcomes and highly favorable opinions regarding learning relevance, interest, motivation, and the recommendation of the course. Therefore, this work contributes to SCM education by including sustainability-related challenges and disciplinary topics in novel instructional designs that will actively prepare future professionals and decision-makers.

**Keywords:** experiential learning; supply chain management; sustainable development goals; educational innovation; higher education

## 1. Introduction

This work relates to the development of sustainability-related learning experiences for the discipline of supply chain management (SCM) in Higher Education. This idea emerges from the contemporary challenges and opportunities that universities face in the type of education they are required to deliver to their students [1–3]. Presently, universities must educate students beyond disciplinary knowledge to develop the right skills to face the requirements for their professional careers and personal development in their corresponding fields and countries [4–6]. In addition, there is a need to engage students with relevant, interesting, and motivating learning activities to improve the effectiveness of their learning performance [7]. Finally, there is a global requirement to contribute to the planet's sustainability in all human endeavors. According to the 2030 Agenda for Sustainable Development [8], education is one of the means of achieving this goal. Therefore, bringing the notion of sustainability to Higher Education and SCM can allow students to learn about real-world contemporary issues close to their personal experiences and impact on their communities and surroundings.

Higher Education must respond to the challenges of humanity by educating students with the required abilities to produce economic prosperity and societal progress, and individually flourish in the world [8]. Some of these challenges include the aftermath of the COVID-19 crisis, the interconnectedness of globalization, the digital transformation of societies and organizations, the future of work, climate change, and the demographic changes in populations [9]. However, one of humankind's most urgent and widespread challenges is sustainability, as defined in the 2030 Agenda for Sustainable Development and the Sustainable Development Goals (SDGs) [10]. In 1987, the United Nations Brundtland Commission defined sustainability as "*meeting the needs of the present without compromising the ability of future generations to meet their own needs*" [11]. Today, there is a global effort to meet the SDGs, but the increasing economic, environmental, and social threats the challenge more significant. Hence, Higher Education must respond to these demands to keep learning relevant for students, organizations, and society now and in the future [12].

In this sense, the notion of sustainability is paramount for Higher Education, as it sets the necessary curricular requirements to educate students in alignment with the existing SDGs [13,14]. Therefore, Higher Education should contribute to sustainability in *SDG #4 Quality Education* to ensure that students acquire the knowledge and skills needed to promote long-term sustainable development [10]. This perspective covers, for instance, incorporating the SDGs and targets into educational models, teaching strategies, learning experiences, and educational resources according to SDG Target 4.7 [10,15]. The aim is to allow graduates to grow sustainability competency in their disciplines of study in a practical and high-impact way.

Thus, sustainability in SCM education should consider the effects that supply chains in organizations have upstream and downstream. It should also consider how these can support or enable sustainable development in communities and their broader environments.

By looking at supply chains as networks that deliver products and services from raw material sources to final consumers through an engineered flow of information, physical distribution, and money [16], we can translate sustainability into strategic, tactical, and operational terms [17]. Therefore, supply chains may generate a favorable footprint of inclusion and equity in cities and their communities beyond economic and environmental aspects. Hence, sustainability requires managing supply chains effectively to achieve expected outcomes.

Following these ideas, studying sustainability challenges in supply chains becomes paramount for education, as it stresses the importance of learning outcomes beyond technical or economic aspects. It also allows moving learning activities outside the classroom and universities to cities, communities, and organizations, changing how we can conceptualize learning activities and spaces [11]. These challenges also represent an opportunity to develop skills in students that benefit their future employability and challenge their status quo to grow competencies [18].

Moreover, from an educational perspective, there are frequent concerns in teaching practice because students seem not to recognize the relevance of their studies and the impact this appreciation has on their learning engagement and career decisions [19]. Thus, the lack of relevancy creates a missing connection between what students learn and the applicability of teaching content to performing current or future jobs or tasks [4]. Relevance to learning also influences the motivation and interest of learners [20,21]. Motivation results from the beliefs and expectations of students about how desirable learning is for them [22]. In addition, the notion of interest describes a durable predisposition of a learner to concentrate or engage with an object or subject over time [22,23]. If there is a shift from situational to personal interest, students increase their chances of engaging in their activities. Thus, interest is a predictor of academic performance [24]. The transition from situational to personal interest rises when someone recognizes learning activities as relevant for future career development and professional practice [22,25].

Teaching in Higher Education should consider meaningful applications beyond textbooks or case studies with the direct participation of students in learning experiences [26–28].

The link between learning and contemporary, relevant topics and hands-on activities is fundamental in improving those experiences [29]. Therefore, sustainability development performance brings a globally relevant context to foster and enhance active learning in SCM education.

According to Lukman et al. [30], existing teaching practices in sustainability and SCM education predominantly use multiple combinations of traditional pedagogical approaches (e.g., lectures, case studies, self-study, projects, problem-based learning, game simulations, and online learning). However, these authors indicate that no work has been conducted on transformational learning approaches, in which students consciously make meaning of what they learn. This gap opens a research possibility for instructional design in disciplinary and educational terms.

Therefore, this work addresses the relevance of learning by studying contemporary challenges in real-world situations concerning the SDGs in SCM education. This approach points to students undertaking purposeful learning experiences to propose solutions to overcome sustainability problems in particular supply chain situations. Thus, this work suggests that this type of learning experience in Higher Education should include (i) highly relevant, interesting, and motivating topics regarding supply chains and sustainability, (ii) educational approaches that create engaging and participatory learning experiences, and (iii) the assessment tools to elucidate the student's views regarding their learning experiences. These ideas can be translated into a research question (RQ) to guide this work, as follows:

> *RQ: How may engaging and participatory learning activities regarding sustainability-related study situations in SCM education enhance students' learning relevance, motivation, and interest, creating highly satisfactory learning experiences?*

This work aims to develop a framework for instructional design to support this effort, exemplify and disseminate its use with a single learning experience case, and contribute to SCM education through an actionable tool. Thus, this work intends to show how the difficulties in providing relevant education, developing relevant skills, and educating in sustainable development can be reduced.

To progress in this direction, this article unfolds as follows. Section 2 sheds light on the relationship between SCM and the economic, environmental, and social aspects of sustainable development for disciplinary and educational purposes. This section also provides the conceptual framework and pedagogical approach to bringing sustainability to SCM education. Thus, Section 2 covers this work's fundamental assumptions and provides the necessary concepts and tools for developing the proposed framework and method. Section 3 presents an application case study to exemplify a learning experience regarding sustainability issues in SCM. Section 4 discusses the results and identifies this work's main findings, limitations, and future work on the topic. Lastly, Section 5 presents the conclusions and contribution of this work.

## 2. Materials and Methods

### 2.1. SCM Education for Sustainable Development: Setting Requirements for Learning Experiences

Adding sustainability principles to Higher Education contributes to sustainable development by educating students to understand the root causes of problems and matching them to suitable solutions [31,32]. Hence, students should approach academic or scientific problems that must raise awareness about the current and future impacts without compromising the possibilities of future generations [33,34].

The United Nations Earth Summit in Rio de Janeiro in 1992 institutionalized the concept of "*Education for Sustainable Development*" [35]. As a result, this declaration prompted various universities to include sustainability topics in their education programs in the following years. These efforts are commonly referred to as *Higher Education for Sustainable Development* [36]. Furthermore, accreditation bodies, such as the *Accreditation Board of Engineering and Technology* (ABET) [37], emphasize the importance of sustainability in their auditing processes and assessment criteria of student learning outcomes (i.e., what a learner

knows, understands, and can do after the completion of learning [38,39]). We can find another example in the *Principles for Responsible Management Education* (PRME) framework, which contains a set of principles to incorporate the SDGs into educational activities and disseminate examples of approaches already adopted by business schools [40]. However, integrating sustainable development in Higher Education is incipient and mostly limited to developed countries [31,41,42].

A systematic change in teaching is required to understand the complexity of contemporary phenomena far beyond that which exists in conventional programs and integrate sustainability principles into Higher Education [41]. Academic programs must creatively integrate sustainability concepts to allow students to understand their importance in daily life and their future professional careers. Most research on this topic justifies the need to clarify the concepts and change curricula. Still, few academic articles have focused on teaching and specified how this change could occur, either at the level of course design, concerning educational methods, or teaching practice [32,36]. The essential efforts in the literature body in this direction highlight the use of mixed methods and resources, covering discussion groups, multimedia, experiments, observation of current events, lectures, role play, project work, debates, question-and-answer sessions, case studies, discussion sessions, assignments, textbooks, and expert speakers [36].

Moreover, to understand the notion of sustainability in SCM education more deeply, we should look at one of the most influential works in business management. The latter highlights the "*Triple Bottom Line*" (TBL) produced by John Elkington. This framework looks at the social, environmental, and economic impacts of human endeavors on the well-being of people, the health of our planet, and the generation of profits and wealth [43]. The TBL favors system change and assesses the social, environmental, and economic performance over time to consider the total cost of doing business [44,45]. In social terms, sustainability applies to the impact that organizations have with their business decisions on multiple stakeholders, such as customers, employees, and community members. Concerning the environment, this is about the contributions to our planet's conservation and climate change. Finally, economic performance points to maximizing profits while reducing costs and mitigating risk. The TBL does not place social and environmental value at the expense of economic profitability, but as a well-balanced commitment to sustainable business practices. Doing well in economic and environmental terms only produces viable benefits. Economic and social contributions only deliver equitable value, and environmental and social contributions limitedly provide bearable value [43].

Regarding SCM education, sustainability is an integral and transdisciplinary concept, and it requires developing skills that consider the TBL and the SDG [32]. Furthermore, future professionals should be conscious of inequalities, decent work conditions, and the need for economic growth, environmental conservation, industrial innovation and infrastructure, responsible production and consumption, and partnerships and alliances to achieve the SDGs and build sustainable supply chains [32,46–49]. Thus, it becomes essential that future supply chain professionals embody and embrace sustainability by participating in convenient learning experiences that equip them with the skills to generate long-lasting solutions in supply chains. However, most efforts in the field refer to environmental impacts (see [50,51]). The latter is a significant limitation to overcome in disciplinary and educational terms, which requires awareness of the consideration of the three aspects of sustainability in SCM education.

Sustainability education in the discipline should relate to existing SCM practices, challenges, and academic curricular requirements [52,53]. It links to topics such as supply chain design, demand planning, logistics involving the supply chain configuration, resource management, supplier management, and the professional code of practice. It can also comprise other notions of logistic operations concerning inventory management, distribution, market management, retail operations, picking, packing, reverse logistics, and waste management [54]. In addition, different sustainability and supply chain learning scenarios might cover diverse stages, such as the first mile, last mile, and intra-city, peri-

urban, intercity, and rural areas [55]. Other features include consumer profiles, product types, demand volume, product price, market segments, logistic infrastructure, vehicle traffic, fuel efficiency, and business sectors [56–59].

A comprehensive framework, shown in Figure 1, integrates all of these ideas for identifying learning experiences regarding supply chains and their sustainability impact in line with the TBL and the SDGs [10,60]. The framework describes materials, information, and money flows alongside segments of supply chains involving different actors, such as manufacturers and producers, distributors and wholesalers, retailers, end-consumers, and logistic operators. The TBL is also incorporated, along with icons representing examples of potential effects on different economic, social, and environmental aspects of sustainability for each situation or circumstance, as previously described.

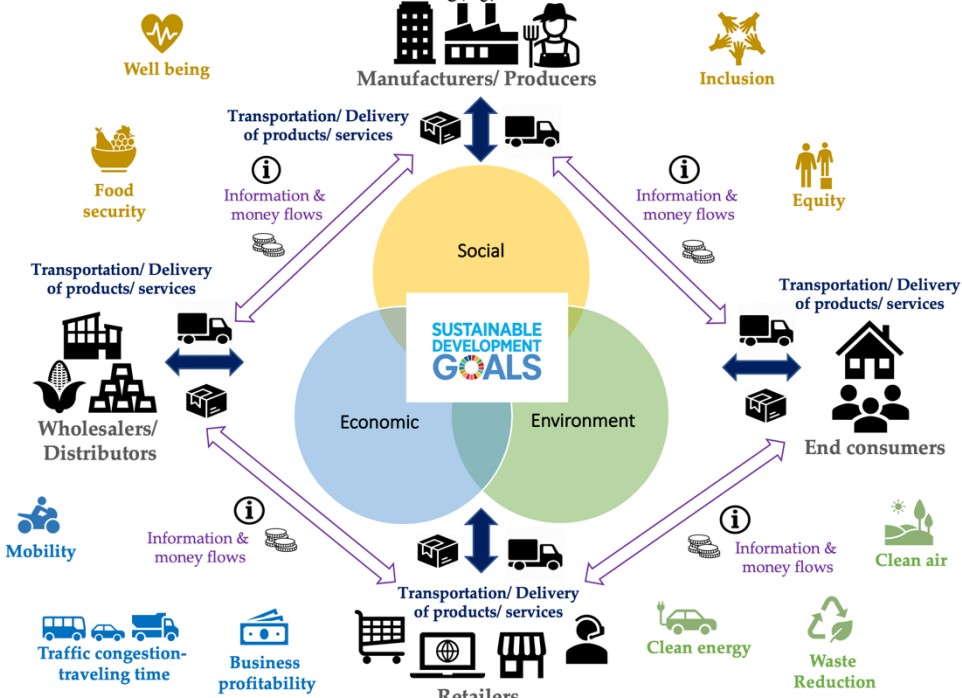

**Figure 1.** A framework of supply chains' impacts on sustainability (own elaboration, adapted from [43,60].

Consequently, in educational terms, this work proposes an alternative to study situations regarding supply chain operations in cities related to *SDG #11 Sustainable Cities and Communities* because of their increasing sustainability impact on the future of humankind. However, future research might consider other options. Urban areas currently account for approximately 60% of the global GDP, and are expected to account for 60% of the worldwide population, 70% of global carbon emissions, and 60% of resource use by 2030 [61]. Moreover, recent trends, including the growth of consumers in urban locations, the rise of e-commerce, digital transformation, and the socioeconomic conditions in emerging countries, have strengthened the focus on urban areas in the twenty-first century [54,62]. There is also the need to look at current challenges in urban logistics, such as navigating traffic parking regulations, the lack of suitable home-delivery infrastructure, the delivery of perishable products to diverse customers, such as a myriad of nanostores, same-day and instant deliveries, distribution visibility, crowdsourcing deliveries, and cost/time efficiency [51]. As a result, several opportunities remain for improving urban supply chain operations in practice and developing educational initiatives in the field.

For instance, supply chain operations in cities can be linked to decarbonization, energy efficiency, waste reduction, improved urban mobility, vehicle accident reduction, land-use impacts, and better reliability and efficiency in terms of operations [17,50]. These

challenges can exist in urban distribution centers, last-mile logistics, cargo (un)loading, and retail operations. Other situations refer to supply disruptions that affect product/service accessibility, availability, and affordability, jeopardizing social inclusion and equity in urban areas [63,64]. Table 1 provides examples of indicators to assess supply chain performance as an alternative to improve the impact of sustainability, providing methods to identify concerns as study situations for conceptualizing learning experiences [65–69]. This view also relates to the targets of *SDG #2 Zero Hunger* with regard to food supply; *SDG #11 Sustainable Cities and Communities* with regard to transport systems, urbanization, air quality, road safety, mobility, and social inclusion; and *SDG #12 Responsible Production and Consumption* concerning the efficient use of natural resources, solid waste generation, and food losses [10]. It is crucial to recognize that these indicators do not precisely match the SDG indicators, as these do not consider specific supply chain situations or their impacts on cities and communities. Nevertheless, a practical and intuitive relationship between supply chains and the SDG targets requires deeper exploration.

**Table 1.** Supply chain and logistics' impact on the sustainability of cities and communities [65–69].

| Supply Chain Performance | Logistics Transport Performance |
|---|---|
| <ul><li>Delivery speed</li><li>On-time and in-full delivery (OTIF)</li><li>Order accuracy</li><li>Fill rate/ service level</li><li>Stockouts</li><li>Asset utilization</li><li>Unit cost per mile, vehicle, or item</li><li>Customer satisfaction</li><li>Customer complaints</li><li>Damage claims</li></ul> | <ul><li>Parking space availability</li><li>Through-freight share of total demand</li><li>Truck utilization</li><li>Netload factor</li><li>Delivery productivity/daily delivery density</li><li>Logistics sprawl (average warehouse distance to customers)</li><li>Driver hours in-motion and inactive</li><li>Routing efficiency (planned vs. actual mileage)</li><li>Number of/time between stops</li><li>Time travel index on freight lanes</li><li>Truck-related casualties</li></ul> |

**Supply Chain Impact on Sustainability Performance**

| Economic | Social |
|---|---|
| <ul><li>Road congestion/mobility</li><li>Circulation speed</li><li>Traveling times</li><li>No-access and no-availability cost and time</li><li>Marginal cost per usage</li></ul> | <ul><li>Traffic accidents</li><li>Level of noise</li><li>Effects on public health/respiratory diseases and level of stress</li><li>Products/services availability, affordability, and accessibility</li><li>Work–life balance</li><li>Job generation</li></ul> |

**Environmental**
- Pollutant emissions/$CO_2$ and suspended particles
- Fossil fuel consumption/efficiency
- Energy consumption/efficiency
- Solid waste generation and recovery
- Land and aggregated infrastructure usage

Incorporating sustainability into SCM education also requires stimulating in students the skills of critical reflection, decision-making, and problem-solving [42,70]. The increasing use in Higher Education of a plethora of teaching and learning approaches, such as competency-based education (CBE), project-oriented learning (POL), problem-based learning (PBL), and challenge-based learning (CBL), among others, can help to strengthen these skills [71–74]. Therefore, teaching and learning activities should use suitable educational methods in their design.

Furthermore, selecting suitable educational approaches requires the identification of a link between study situations and the type of engagements students can undertake in their modules or courses. That is, study situations can use fully immersive scenarios from the real world or just informative setups related to specific problems. In contrast, learning experiences can be face-to-face in a classroom, hybrid, or remote over web-based sessions,

synchronous in real-time meetings, or occur in asynchronous virtual environments, which require creating suitable instructional designs [75]. Hence, there is a task to advance learning experiences under the previous requirements and criteria.

A learning experience relates to "*the specific engagements of students and teachers in their everyday lives, their activities, and their social interactions in real-life settings and contexts, in classrooms and beyond, with learning purposes*" [76]. Learning experiences transform learners' perceptions, facilitate conceptual understanding, induce emotional qualities, and promote the acquisition/transfer of knowledge, skills, and attitudes. Moreover, learning experiences are ideally challenging, engaging, and meaningful to meet learners' needs, becoming key factors that further improve education performance [77]. The notion of a learning experience is regarded in this work as the fundamental unit of analysis or *research object* [78].

Sustainability-related learning experiences for SCM education require a transdisciplinary approach to increasing students' awareness, modern management techniques, businesses and community outreach, and the formation of job-ready skills [79]. This idea concerns relevant and novel learning experiences to produce high-impact and long-lasting learning in future generations of SCM professionals [26]. These experiences provide students with immersion in real-world contemporary study situations, rather than closed descriptions, examples of past events, or abstract situations from non-relevant contexts.

*2.2. Experiential Learning in SCM Education for Sustainable Development: Translating Theory into Educational Practice Using Instructional Design Tools*

In the education-related literature on SCM, some authors recognize that relevance points to creating a suitable curriculum, updating its content, and creating instrumental educational resources [80–82]. Other scholars emphasize using innovative teaching methods to integrate collaborative practices, workshops, gamification, field trips, and guest speakers [53,83,84]. Others stress the need to make advancements in developing appropriate skills and competencies to meet industrial requirements by reviewing teaching content and assessment methods, and identifying gaps for curriculum improvement [85]. Finally, SCM education should promote students' active participation and interaction in developing their learning outcomes [27,72,86–88]. Therefore, learning relevance links to *what to teach or learn* and *how to teach or learn*.

This work considers the development of learning experiences to study relevant sustainability challenges in real-world supply chains in line with the SDGs. It becomes paramount to provide disciplinary authority and alignment with the global effort in this field. This view focuses on *what to teach or learn*. Additionally, the learning experiences should involve student-centered and collaborative work in purposeful, situated activities that produce long-lasting and impactful learning through improved engagement [89,90]. This proposition points to *how to teach or learn*. Nevertheless, there is little work in the SCM education literature demonstrating how to undertake this effort (see for details [52]). Consequently, significant changes in learning experiences for SCM education should occur to prepare students for their future professional challenges [79].

If high-impact learning is required, *Experiential Learning* reinforces students' motivation to learn and their long-term retention through practicality [91,92]. Approaching learning from this perspective requires a different educational action-oriented framework to conceptualize, organize, and implement meaningful learning experiences in real-life environments that assist students in conducting reflective practice, decision-making, and problem-solving approaches [15,93]. In addition, this approach provides students with an active role and responsibility for their learning with the support and mentoring of academics and continuous feedback to students on their progress [94].

Experiential learning is a constructivist theory of learning that emphasizes what students must do to construct knowledge. It suggests the types of learning activities teachers need to encourage students to perform to achieve their intended learning outcomes [95]. Therefore, from this view, teaching is not about broadcasting information, but engag-

ing students in active learning, building their knowledge in terms of what they already understand.

Experiential learning contains an integrated four-stage process composed of observation, data collection, analysis, and elaboration of conclusions, which contributes to the modification of behaviors and selection of new experiences [96]. This type of learning considers producing a recursive circle of a concrete experience (CE), reflective observation (RO), abstract conceptualization (AC), and active experimentation (AE), which naturally occur in a continuous meaning-making process loop [97,98]. CE refers to a new experience or situation that triggers a stimulus to actively engage in a task, rather than merely reading or watching. RO is about reflecting on the new experience, recognizing any possible discrepancies and gaps between the learner's understanding and the experience. AC concerns new ideas or modified thoughts coming out from the reflection. It also includes interpreting and updating experiences from new knowledge. Finally, AE refers to what the learner applies to the outer world. It is also known as the testing stage to apply conclusions to new experiences [96,99,100].

Sivalingam and Yunus [101] proposed a link between the stages of the experiential learning cycle and Bloom's taxonomy levels [102,103] concerning student learning outcomes. In contrast, Bloom's taxonomy supports the definition of educational objectives and the level of expertise required to achieve each measurable student outcome. Accordingly, CE relates to applying RO to analyze, AC to create, and AE to evaluate. However, this work focuses on conceptualizing learning experiences in their activities under predefined sustainability-related learning objectives and outcomes.

Additionally, experiential learning emphasizes taking learners to situations in which they can learn from an iterative cycle process about, for instance, problem-solving or decision-making. This approach involves covering situational observations, problem assessment, solution design, and validation, which increase students' capacity for effective action in a contextual situation [104]. Each cycle stage depends on its predecessor and follows a continuous logical pattern step-by-step. According to Kolb [99], learning spontaneously occurs as part of a continuous meaning-making process through personal and environmental experiences in which the learner experiences, reflects, thinks, and acts in a situation. Accordingly, experiential learning involves defining and organizing learning activities following the recursive cycle.

Some arguments against experiential learning arise from claims of insufficient attention to cultural differences, the contextual conditions of learners and educators, people's emotions, learning modes, learner types, learning styles, how learning processes connect to knowledge acquisition, and whether learning occurs in identifiable stages [105]. However, scholars also recognize its popularity and wide use in teaching practice [106].

Figure 2 presents the integration of sustainability and supply chain topics into the *Experiential Learning Cycle*. Referring to SCM in cities, CE relates to students perceiving and collecting data from peri-urban, intra-city, and last-mile operations in streets and neighborhoods. RO relates to students thinking about the implications of supply chain operations in the livelihood of communities, natural conservation, and wealth generation. The latter can use different disciplinary and sustainability-related frameworks to assess results. AC involves students in disciplinary problem-solving, decision-making, and developing solutions to tackle sustainability problems. Finally, AE concerns students planning and implementing their proposed solutions in SCM. The four stages represent a circular interplay between thinking/conceptualizing and experiencing/acting in a situation, transforming the grounding of ideas into effective actions and behaviors [104]. Students should cover the four stages to complete the experiential learning cycle.

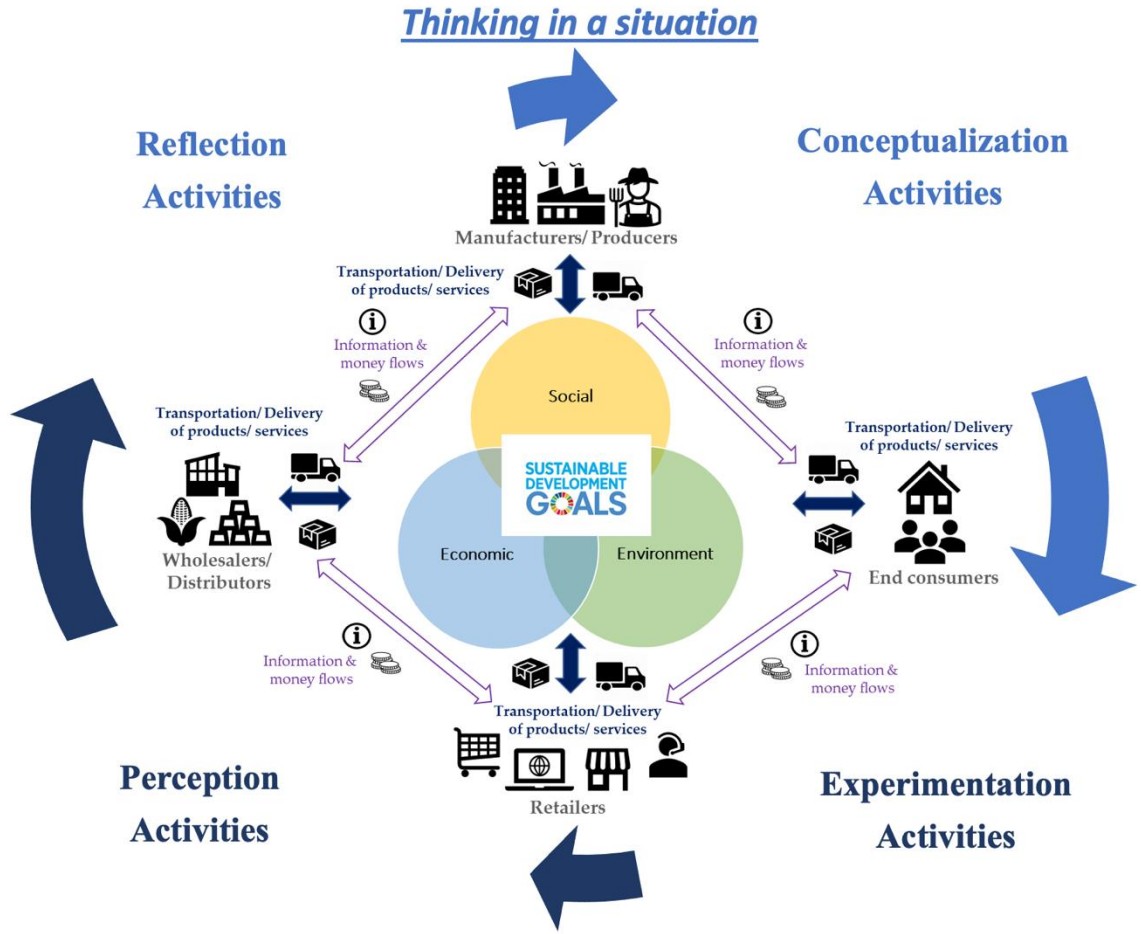

**Figure 2.** Experiential learning cycle for sustainability in SCM education (own elaboration, adapted from [43,96]).

Following these ideas, this work looks at learning experiences based on experiential learning to achieve educational objectives, cover specific SCM academic content, and develop disciplinary and sustainability-related learning outcomes. Experiential learning can support study situations where face-to-face or remote, immersive or informative, and synchronous or asynchronous learning experiences occur in different learning spaces or environments [75]. For instance, this is the case for in-person, online, blended, or hybrid instructional formats. Thus, the learning experiences may include sets of activities following the experiential learning cycle under different learning environments.

Nevertheless, these learning experiences require an instructional design to devise their specific activities. Therefore, this work regards instructional design as "*the systematic development of a delivery system using learning and instructional theory to identify and meet learning needs and goals*" [107]. In this sense, the ADDIE model for instructional design can guide the conceptualization of learning experiences concerning their analysis, design, development, implementation, and evaluation, as presented in Table 2 [108,109].

**Table 2.** ADDIE model-based framework for learning experience documentation [108,109].

| | |
|---|---|
| Analysis | 1. Module/Course selection |
| | • Choose a module or course covering SCM topics in Higher Education; <br> • Address the existing educational model requirements. |
| | 2. Problem situation/challenge definition |
| | Select real-world situations concerning the impact of supply chain operations on the sustainability of cities and their communities in line with the SDGs as learning challenges or problem situations. |
| | 3. Disciplinary learning objectives |
| | Define learning objectives about supply chains and their sustainability impact. |
| | 4. Learning outcomes and competencies |
| | Determine disciplinary learning outcomes and competencies regarding: <br> • The design and development of sustainable solutions for SCM; <br> • Ethical commitment and citizenship—Ethical commitment to social transformation. |
| | 5. Format |
| | Select instructional formats for: <br> • Immersive or informative situations; <br> • Face-to-face, online, blended, or hybrid learning spaces. |
| | 6. Target learners |
| | Set instructional formats based on the study level, academic discipline, and academic program. |
| Design | 7. Knowledge acquisition |
| | Define disciplinary topics in SCM. |
| | 8. Teaching and learning approach/strategy |
| | Choose Experiential Learning as the leading instructional approach. |
| | 9. (Experiential) Learning activities |
| | Design and describe the learning activities in the Experiential Learning Cycle. |
| Development | 10. Educational resources |
| | Prepare educational resources and materials for the course/module. |
| Implementation | 11. Course/Module execution |
| | Carry out the learning experience through lectures, seminars, and other interactions. |
| Evaluation | 12. Learning outcomes and experience evaluation |
| | • Provide coursework briefs, rubrics, and student evaluation instruments; <br> • Conduct student surveys at the beginning and the end of the course. |

The ADDIE model provides high-level guidance for developing and revising instructional designs [110]. The ADDIE model refers to an instructional iterative design process with dedicated stages, representing a common standard approach widely used in developing instructional courses and training programs [111]. Following these ideas, using this model emphasizes the learner, rather than a teacher-centered approach, as it provides a process that actively engages students in their learning activities (e.g., problem-solving, decision-making, or policymaking). The process can be applied to various settings because of its systematic and generic structure. The framework provides designers with a means for identifying the target audience's needs and reinforces the use of this information for the design and development of programs. Throughout the implementation phase, developers employ the ADDIE model to gather the necessary feedback (through formative and summative evaluations) to determine the program's effectiveness. The developer then takes corrective action or makes the changes and adaptations necessary to deliver the program successfully. The evaluation stage ensures that the target audience's needs are met [108–111].

The analysis stage relates to the definition and selection of the problem situation or the definition of challenges, learning objectives and outcomes, format, course or module organization, and the target group of learners. The design stage involves students' knowledge acquisition, the teaching and learning approach or strategy, and the learning activities. The development stage is related to the educational resources, including the syllabus, learning spaces and environments, learning activity map, instructional materials, and other educational resources. The implementation stage involves the communications and interactions teachers and students have throughout the learning experience, either face-to-face, remote, online, blended, or hybrid, in their learning spaces or environments, such as lectures, seminars, tutoring sessions, or other types of interactions. Finally, the evaluation stage covers the formative and summative evaluations of students' learning achievements. It also might include the assessment of students' expectations, the relevance they grant to their knowledge, their interest and motivation, their attitude development, and their level of engagement, among others. The five stages communicate back and forth and provide feedback to maintain coherence, (re)shaping the conceptualization and implementation of a learning experience. Table 2 exhibits an adapted ADDIE model-based framework for documenting learning experiences in these terms, considering sustainability and SCM education requirements.

The adapted ADDIE framework is a tool to conceptualize learning experiences in instructional designs. However, there is still the need to create additional tools to develop specific resources not covered in this work, such as syllabus templates, learning outcomes maps, student journeys, activity calendars, assignment briefs, exams, assessment rubrics, and surveys [112].

The ADDIE framework can be applied to diverse engineering and management courses or modules covering SCM and sustainability-related topics to recreate experiential learning activities. This work proposes using the *continuous improvement cycle*, namely *Plan–Do–Check–Act*, or the *PDCA Cycle*, to define a method that guides its implementation (see Table 3) [113,114]. From this cycle, the *Plan* stage helps to identify an opportunity and designs for action. *Do* is related to deploying and undertaking actions. *Check* involves analyzing the results and determining if they are satisfactory or not. Finally, *Act* assesses if the actions were successful and whether implementation should happen on a broader scale, and continuously assesses the results.

**Table 3.** PDCA Cycle for ADDIE [113,114].

| PDCA | Steps |
|---|---|
| Plan | Use the ADDIE framework to develop an instructional design. |
| Do | Execute the instructional design as a learning experience. |
| Check | Collect observations, assess the student learning experience, and reflect upon results to improve further instances. |
| Act | Produce concluding statements and feedback and develop changes to achieve expected results, if necessary. |

The ADDIE model-based framework within the PDCA cycle is later exemplified in this work in an exploratory case study of a learning experience in an undergraduate program (see Section 3). The framework is used to structure and describe this learning experience. Thus, this case describes the elements of a learning experience and the analysis of a single instance according to the PDCA Cycle in Table 3.

### 2.3. Research Methodology

This work proposes an instructional design framework related to the RQ to develop learning experiences involving (i) highly relevant, interesting, and motivating topics regarding supply chains and sustainability, (ii) educational approaches that create engaging

and participatory learning experiences for Higher Education, and (iii) the assessment tools to elucidate the student's views regarding their learning experience (see Section 1).

To progress in this direction, a five-step methodology was devised based on the ideas of Popper [115], De Zeeuw [78], Vahl [116], and Tharenou et al. [117] on conducting research in the social domain, as follows.

1.  Define what to observe relative to the RQ;
2.  Choose the research design and select an instance of the research object (i.e., a learning experience);
3.  Collect data and construct formulations and statements relating to the research object;
4.  Evaluate and interpret results against the research object and redefine or discard statements and claims, if necessary;
5.  Report the findings and decide on further action by using the results of step 3.

In step 1, an RQ is proposed in Section 1 to define what is relevant to the research aim of this work. This idea refers to the underlying theories and frameworks supporting this work (see Section 2.1 and 2.2). These concepts allow for proposing a unit of analysis in Section 2.1 as a research object, i.e., a *learning experience*.

Referring to step 2, this work's research design considers an exploratory single case study to advance in answering the RQ (see Section 3). The case contemplates a learning experience linked to a unique situation, location, group of people, or event to explain and gain insight into its particularities, rather than other cases or generic issues [118] (pp. 62–64) [119]. The case study illustrates the instructional design of a learning experience in an undergraduate program using in-depth exploration based on the ADDIE model-based framework (see the next step). A case study is selected in this work as it applies to unique situations or explains the implementation of new methods and techniques where there is only one or a small number of situations or instances. Therefore, no comparisons are made with control groups to develop inferences or generalizations about other instances or situations [119].

Concerning step 3, a mixed-methods approach for data collection and analysis helps to construct formulations and statements regarding the research object (see Section 3.2.3). Observational reports from instructors (i.e., two authors) regarding their course instructional design and the learning experience are collected as primary data. These data provide the necessary background information based on the ADDIE framework. Additionally, secondary data are collected regarding students' examinations to provide information about the numeric evaluations of their learning results (formative or summative), such as exams and reports, an assessment of disciplinary and sustainability-related learning outcomes, and an assessment of student opinions regarding the course and the learning experience. Some of these data (i.e., student opinions on the learning experience) are collected through a longitudinal process with an intervening period during an academic term. Later, the collected data are analyzed using descriptive statistics (i.e., mean, standard deviation, median, and interquartile range) and non-parametric tests (i.e., Mann–Whitney test) to describe and elaborate on formulations and statements regarding the learning experience's outcomes.

Step 4 discusses results against the underlying theories and frameworks (i.e., learning experiences, experiential learning, ADDIE model, and learning outcomes), the research object, and the RQ (see Section 3). If results from the data analysis are unsuccessful, that is, the results differ from the supporting theories and/or claims regarding the research object, these statements will require redefinition (or being discarded) or the implementation of further actions (i.e., improvements) as defined in the PDCA cycle.

Finally, in step 5, research findings are presented, including limitations and future work on further instances of the research object, which may require going back to step 2 in a continuous cycle (see Section 4). If claims, formulations, and statements regarding the research object achieve stability (i.e., do not change or vary) over further instances of the object, the results might be transferred, applied, or used to improve on other instances (by other researchers). As this work focuses on the social domain because of the study

of learning experiences where students, academics, and educational partners engage, the evaluation of the research results requires the criteria of *reliability*, *transferability*, and *validity* [116]. Therefore, *reliability* means whether the collected observations are repeatable or consistently attributed to instances of the same object (i.e., learning experiences). *Transferability* indicates if (other) researchers can identify new occurrences of the object and where they can consistently use observations without modification, achieving *observational closure*. Finally, *validity* raises the question of how confident they are in how the interpretation of observations can always refer to the same object in reality. These criteria will guide discussions regarding the case study to identify implications and future work.

Moreover, *observations* relate to data, opinions, or reports of what people claim to have seen or experienced [78]. Observations are also *observed-dependent,* given that they depend on the viewpoint, behavior, or reactions of what or who is being observed (e.g., students and academics) [78,116]. Hence, the validity of observations, their analysis, and their interpretations are constrained to single instances of a learning experience.

In summary, the presented methodology concerns a description of the research object, research design, the type of observations to collect, a framework to design learning experiences, an actionable method to guide the research process, and criteria for the research results' evaluation. Accordingly, the following section presents an application of the methodology to describe its implementation and collect data regarding one instance of a learning experience.

## 3. Results

This section describes a learning experience as an application case study, referring to the method offered in Table 3. The application case unfolds in two subsections. First, the background situation of an undergraduate course, its justification, its educational implications for SCM education, and its relation to sustainable development are presented. Second, an application of the method in a particular instance is reported, describing *what* and *how to learn* about sustainability for SCM education.

### 3.1. Background Situation

The IN2005 System Dynamics course at Tecnologico de Monterrey University on the Mexico City Campus involved the creation of novel learning experiences over the last few years about the impact of supply chains on the sustainability of cities and metropolitan areas. This course is part of the seventh semester in the Industrial and Systems Engineering (IIS in Spanish) undergraduate program version 2011. The School of Engineering and Sciences offers this program across 26 campuses of this private, non-profit university. Therefore, the design of these learning experiences for the Mexico City Campus was in line with the institutional requirements at the university.

IN2005 System Dynamics should develop fundamental systems-thinking skills in providing system-as-a-cause explanations, dynamic behaviors, 10,000 m-altitude thinking, operational thinking, generic thinking, causal-loop and stock-and-flow modeling, emphatic–ethical thinking, and observer-dependent viewpoints [120]. Students should also apply these skills for problem-solving and policymaking in complex situations. According to the institutional mission, the course should also contribute to students' education in developing citizenship and ethical outcomes [93].

In disciplinary terms, IN2005 System Dynamics explores sustainability issues related to dynamic complexity. It examines phenomena, events, patterns of behavior, system structures of feedback loops, and mental models to identify leverage points for policy-making, as initially explored in the book *Limits to Growth* by Donella Meadows et al. in 1972 [121,122]. Furthermore, system dynamics also provides the concepts and tools to study supply chains concerning the bullwhip effect and inventory oscillations deeply rooted in poor decision-making and structural deficiencies in information and material flows [123,124]. Thus, exploring sustainable development issues in SCM from the perspective of system dynamics provides an excellent opportunity for expanding their study and enhancing their valuable

contribution to Higher Education. However, according to Tobias et al. [125], existing works and resources for system dynamics concerning SCM and sustainability are scarce and mostly focus on climate change and environmental issues (see also [126–128]). Thus, this limitation makes it necessary to explore different topics to incorporate into SCM and system dynamics education.

Specifically, IN2005 was selected as a case study for educational innovation in SCM and sustainability because of the impact that the COVID-19 pandemic had on food chains in terms of SDGs #2 Zero Hunger, #11 Sustainable Cities and Communities, and #12 Responsible Consumption and Production, and food security. At the dawn of the COVID-19 crisis, food supply and demand showed behavior patterns corresponding to inventory oscillations and the bullwhip effect [123]. That is, food demand suddenly increased because people were concerned about food availability, creating a hoarding effect that rapidly led to inventory stockouts. Conversely, food supply was disrupted or interrupted because of sanitary restrictions and social distancing, increasing food distribution and delivery times to consumers. As a result, significant demand and supply gaps developed in a short time, creating inventory oscillations and reverberations in food stock levels over time because of consumers' and logistic operators' panic decisions, inadequate resupply, and limited food availability and accessibility. These difficulties in food supply were widely experienced by people, causing food shortages and affecting food security in countries such as Bolivia, Colombia, Mexico, and Peru [129]. The COVID-19 pandemic allowed these effects beyond sanitary and medical aspects to be studied to assert SCM as an essential discipline during the sanitary crisis. Thus, system dynamics could help to define supply chain strategies across the retail landscape to support food security in metropolitan areas during the COVID-19 crisis.

Moreover, the course was associated with the *Social Lab for Sustainable Logistics* (SLSL), an educational innovation initiative to explore the sustainability of complex issues in supply chains to improve student learning relevance, interest, and motivation [130]. In addition, a partnership started in 2016 with the *Food and Retail Operations Lab* (FaROL) at the MIT Center for Transportation and Logistics (CTL). This collaboration helped to provide students with learning experiences that could integrate topics such as retail operations, SCM for nanostores (e.g., corner shops and "mom and pop" stores), and food security in metropolitan areas in emerging market economies (see [131]). Over the years, different learning experiences have occurred in this collaboration about the following topics:

- The impact of social and cultural issues on store choice in the metropolitan areas of emerging markets;
- The contribution of supply chains to the sustainable development of neighborhoods in the metropolitan areas of Latin America and the Caribbean to improve the daily lives of citizens;
- Overcoming barriers to improving food supply in neighborhoods over the COVID-19 pandemic;
- Supply chain strategies to combat malnutrition through nanostores;
- Cash-constraint operations in nanostores.

Students approached these situations over their course by applying their disciplinary knowledge of system dynamics, supply chains, and sustainability to study a problem situation and produce a final research report. Students also participated in seminars to capture the purpose and details of their undertaking, receive training, and learn about the sampling and data collection protocol. Other seminars offered a review of the background situation, existing work in the field, challenges, and future work, and provided follow-ups to students, clarified doubts, and answered questions. The execution of the learning experiences resulted in highly creative works in which students identified problem situations close to the reality of their neighborhoods and the organizations in which some worked. For instance, students developed the following research works:

- The contribution of nanostores to obesity and food malnutrition in Mexico City;

- The support nanostores provide to local producers for sustainable neighborhood development;
- Increasing the competitiveness of nanostore business models for different socioeconomic levels;
- Nanostore supply chain strategies to overcome the competition among convenience stores and supermarkets in urban retail landscapes;
- Nanostore strategies for reducing waste generation in neighborhoods.

These learning experiences progressed over the years in their conceptualization, implementation, and the development of an actionable educational framework presented in this work. It must be mentioned that other complementary collaborations occurred to enrich the students' learning experiences. This was the case of the participation of eight high-performing students in research stays at the Eindhoven University of Technology (TU/e) in the Netherlands, the MIT Center for Transportation and Logistics (CTL), the Social System Design Lab at Washington University in St. Louis, USA, and the Centro Latinoamericano de Innovacion en Logistica (CLI) in Colombia. Additionally, these students contributed four works to the student paper competition at the MIT SCALE Latin America and the Caribbean Conference.

Additional collaborations with academics arose to improve and roll out similar learning experiences in other modules and disciplines. One example involves a biomedical engineering course at Tecnologico de Monterrey that allowed students to explore novel scenarios and learning spaces where supply chain operations affect health conditions and people's well-being (e.g., logistic operators, staff, and citizens). There was also the case of collaborations with universities in the MIT SCALE network for Latin America and the Caribbean in Bolivia, Colombia, Mexico, and Peru to design and implement learning experiences for industrial engineering education considering sustainability challenges for local communities, private companies, and organizations [132].

Therefore, the evolution of the conceptualization of learning experiences for IN2005 System Dynamics resulted from the existing situation created by the COVID-19 pandemic and a fruitful collaboration toward innovation in SCM education. The learning experience described here is about the impact of supply chains on the food security of neighborhoods in Mexico City over the COVID-19 pandemic crisis in 2021. Specifically, this instance considers the ADDIE framework in Table 2 following the previous work of testing and documenting the tool. The following section describes the design of this instance of a learning experience.

*3.2. Applying the PDCA Cycle for the ADDIE Model-Based Framework*

3.2.1. Plan Stage

1.  Analysis—Module/course selection.

IN2005 System Dynamics is an intermediate course that provides students with the fundamental and intermediate concepts of system dynamics, focusing on their applications to industry and society. IN2005 recommends the use of problem-based learning as a pedagogical strategy. Moreover, this course contributes to the ABET accreditation process and the development of engineering student outcomes for undergraduate programs [37]. Finally, this course also incorporates transdisciplinary learning outcomes according to the university's social education program [133], seeking students to:

- Know and be sensitive to social, economic, political, and environmental realities;
- Act with solidarity and citizen responsibility to improve the quality of life in their communities.

2.  Analysis—Identify relevant sustainability issues of concern, problems, or challenges in cities and their communities concerning supply chains.

The design of a new learning experience and its adaptation to the circumstances of the COVID-19 crisis required significant changes in 2021 compared with previous efforts in this course. The pandemic crisis required accommodating social distancing and sanitary

restrictions into a new design and type of study situation students could be approaching. Accordingly, the COVID-19 sanitary emergency emerged as a study subject, as it has implications beyond medical issues that affect all endeavors of humankind. In addition, the pandemic provided an opportunity to learn and explore new ways to face existing and future challenges in supply chains to tackle product and service accessibility, availability, and affordability problems.

An instance of this situation involves the inefficiencies in food systems during the pandemic, which jeopardized food security and sustainability [129]. Cities and communities suffered from problems in obtaining food because of interruptions, barriers, and limitations, especially in developing countries and underserved communities. News reports about consumers' hoarding, bullwhip inventory fluctuations, inconsistent inventory replenishment decisions, or long delivery times in last-mile operations are examples of this problem [63]. Supermarkets, corner shops, city markets, street vendors, and even food producers offered alternatives with different results for customers at their service level. Preparedness in terms of food security requires a holistic and inclusive approach involving diverse actors and their collaboration to strengthen national, regional, and even local food systems [134]. However, underserved and low-income populations often suffered from a lack of food throughout the pandemic, as indicated in various news media reports [41,86]. As the COVID-19 pandemic has strongly affected education because of changes in instructional requirements and learning activities, further adaptations to current educational activities should exist to keep learning experiences according to the existing pandemic challenges in societies. Therefore, IN2005 System Dynamics is a specific instance of these adaptations through collaboration with FaROL and the SLSL to study sustainability and SCM topics in cities and metropolitan areas in Latin America and the Caribbean.

3.  Analysis—Learning objectives.

The IN2025 System Dynamics course aims to simulate, validate, and sensitize diverse complex scenarios or situations using specific system dynamics software. Upon completion of this course, students should use basic systems thinking and system dynamics concepts and tools to study an organizational or social process through model development, implementation, validation, and maintenance. The learning content comprises students learning causal-loop modeling, systemic archetypes, stock and flow models, applications concerning innovation adoption, population dynamics, supply chain (re)design (to avoid inventory oscillations and the bullwhip effect), and infectious disease propagation. Moreover, the objective extends to applying system dynamics by incorporating a learning experience regarding the impact of food value chains on food security. This can help students to understand the dynamics of household food supply throughout the pandemic by exploring the complex causal relationships and effects in the situation to identify feasible alternatives for policymaking to strengthen sustainability and food security in the neighborhoods of Mexico City.

4.  Analysis—Learning outcomes.

The definition of ABET disciplinary student learning outcomes (H and K) and citizenship and ethical commitment transdisciplinary outcomes are as follows:

- Learning outcome (H) is "the broad education necessary to understand the impact of engineering solutions in a global, economic, environmental, and societal context" [37];
- Learning outcome (K) is "an ability to use the techniques, skills, and modern engineering tools necessary for engineering practice" [37];
- Learning outcome *citizenship commitment to social transformation* is "an ability to create committed, sustainable and supportive solutions to social problems and needs through strategies that strengthen the common good" [133].

5.  Analysis—Format.

The learning experience requires an immersive study where students observe and collect primary data directly from households (i.e., family members, relatives, friends, and

neighbors) about their opinions on their food supply experience, before and during the COVID-19 pandemic, in the Mexico City metropolitan area. Owing to the sanitary conditions and limitations, the delivery of course lectures, seminars, tutoring, and collaborative work among students was conducted in remote (online) synchronous sessions.

6.  Analysis—Target learners.

The target learners are IIS undergraduate students in their seventh semester.

7.  Design—Knowledge acquisition.

According to the learning objectives, the learning experience covered the following disciplinary topics:

- Fundamental system dynamics concepts to address environmental, social, and organizational situations (see [122]);
- System dynamics modeling (causal-loop, systemic archetypes, and stock-and-flow) and leverage-point identification for policymaking (see [135]);
- The bullwhip effect and inventory oscillations to understand the effects that delays, decision-making, supply chain structure, and demand patterns/consumer behavior have on inventory levels and stock availability (see [123,136,137]).

8.  Design—Teaching and learning approach.

Collaborative learning complements experiential learning to develop individual and collective learning activities (see [138]). This approach also considers formative and summative learning evaluations of learning outcomes (see [139–141]).

9.  Design—(Experiential) learning activities.

The learning experience intends for students to carry out their activities according to the Experiential Learning Cycle. Table 4 summarizes the experiential learning activities. These activities combine disciplinary study activities related to learning system dynamics content and those corresponding to the problem situation involving synchronous and asynchronous individual and collaborative work. Figure 3 shows a graphical description of the learning experience regarding the impact of supply chains on food security and sustainability in urban contexts involving experiential learning activities. This figure integrates a conceptualization of food chains; the effects on the final consumers' food security; their awareness of the accessibility, availability, and affordability of products; the effects on food quality and delivery times; and the immersive exploration of the food supply over the pandemic by students learning from home. These activities were also mapped onto the system dynamics method for problem-solving, guiding students' disciplinary learning during the course, referring to the experiential learning cycle, as summarized and shown in Table 4 [124].

10.  Development—Educational resources.

This course required educational resources involving:

- A syllabus based on an institutional template informing students about the learning objectives, learning outcomes, content, learning activities, assessment criteria, learning materials, a reading list, and a bibliography;
- A web-based learning platform in Canvas © and Zoom © to facilitate webinar sessions, remote mentoring, and virtual collaborative work;
- Household scenarios as learning spaces to explore food supply issues during the pandemic;
- System dynamics, SCM, sustainability, food security slide packs, and reading lists;
- Vensim PLE © system dynamics modeling software.

**Table 4.** Experiential learning stages and activities.

| Experiential Learning (Bloom's Taxonomy Level) [96] | Activities Description [122,124,135] | System Dynamics Method (Steps) [124] | Type of Activity [138] |
|---|---|---|---|
| Concrete experience (Apply) | • Collect and tabulate quantitative data regarding food supply at home with family members and acquaintances during the COVID-19 crisis using a pre-designed survey over social networks;<br>• Collect and classify qualitative data (i.e., observations and reports) about household food supply practices over time regarding product assortment, quality, delivery times, accessibility, availability, and affordability of food items;<br>• Examine key variables affecting food supply practices concerning order size, purchase amount, product categories, retail format and location, delivery times, service times, service level, and household location;<br>• Plot reference mode graphs. | 1. Problem articulation definition. | Individual work. |
| Reflective observation (Analyze) | • Analyze the aggregated survey database using descriptive and inferential statistics to identify the variables' patterns, correlations, and relationships;<br>• Diagnose a problem or issue of concern about food supply during the pandemic regarding the quality, delivery times, accessibility, availability, and affordability of food items;<br>• Relate the problem or issue of concern to system dynamics theory, causal relations, archetypes, basic structures, and application models to identify similarities, invariances, and relations. | 2. Dynamic hypothesis formulation (part A). | Individual and collaborative synchronous and asynchronous work. |
| Abstract conceptualization (Create) | • Formulate hypotheses about the situation involving food supply and demand;<br>• Elaborate causal-loop models and systemic archetypes to explore the situation's complexity;<br>• Discover peoples' beliefs, values, and viewpoints on the situation. Then, define model specifications, estimations, and consistency;<br>• Compare models with reference modes to match behaviors and systemic structures. | 2. Dynamic hypothesis formulation (part B).<br>3. Model elaboration<br>4. Model testing and validation | Collaborative synchronous work. |
| Active experimentation (Evaluate) | • Evaluate leverage points for a systemic intervention to improve the food supply;<br>• Summarize and defend a proposal based on the leverage points to overcome the situation;<br>• Write up a research report. | 5. Policy design and evaluation | Individual and collaborative synchronous and asynchronous work. |

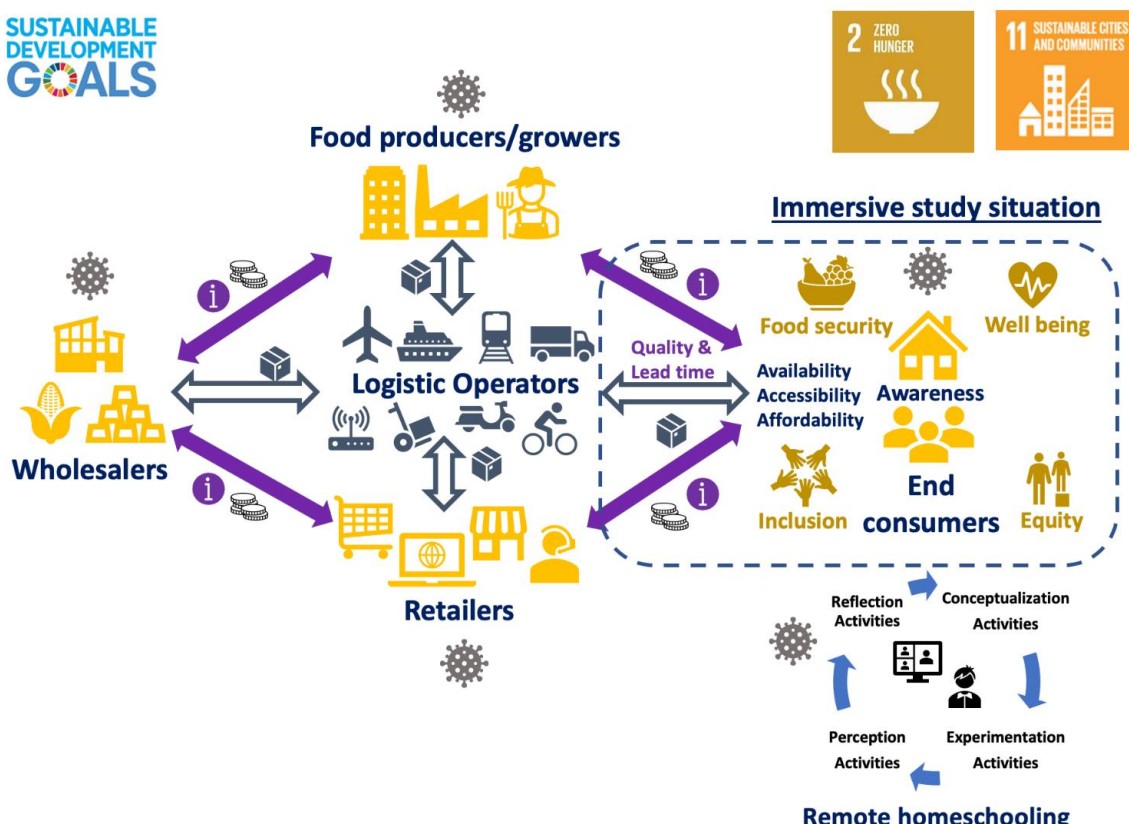

**Figure 3.** Experiential learning on the impact of supply chains on food security and sustainability (own elaboration, adapted from [10,60,96]).

11. Implementation—Course/module execution.

This course's execution of a learning experience occurred over sixteen weeks throughout a semester term. The course involved six hours of teaching webinar sessions plus three open hours for on-demand mentoring. In addition, students were to conduct five hours of independent individual and collaborative work per week. Accordingly, five out of thirty-two sessions covered:

- An introduction to the study situation, justification, objective, structure, assessment, and learning outcomes;
- An exploration of the study situation in the real world, where food supply chains impact food security and sustainability in cities. This session primarily aims at concluding with the concrete experience (CE) stage of students carrying out the Experiential Learning Cycle;
- A presentation and discussion of relevant system dynamics and SCM work addressing critical aspects of the issue. This session relates to the reflective observation (RO) stage;
- A discussion of alternatives to overcome the problem situation based on system dynamics and SCM concepts, methods, and tools. This session focuses on abstract conceptualization (AC);
- A presentation and discussion of students' proposals, implications, limitations, and future work. This session features active experimentation (AE).

12. Learning outcomes and experience evaluation.

The learning experience covers three categories of evaluations. First, evaluations of students' learning results (formative or summative) via exams and reports. Second, an assessment of learning outcomes referring to ABET (H and K) and citizenship commitment. Third, an assessment of student opinions on the course and the learning experience. The following are the specific evaluations and assessments.

- Evaluations (Formative and summative);
  - ○ Two partial exams and one final exam for summative evaluations;
  - ○ Two project partial reports as formative evaluations;
  - ○ A project report for a summative evaluation.
- Assessments of student learning outcomes (disciplinary and transdisciplinary);
- Evaluations and assessments of the student learning experience.
  - ○ Surveys (initial and final) about the student learning interest, motivation, and relevance in the learning experience;
  - ○ An institutional student opinion survey concerning the teaching methodology, academic support, evaluation, feedback, and course recommendation, among others.

### 3.2.2. Do Stage

An instance of a learning experience for IN2025 System Dynamics occurred during the 2021 spring term, covering the design elements in the previous plan stage. The execution of the learning experience required the following considerations as part of the research work.

1. About the IN2005 System Dynamics course

The learning experience involved sixteen IIS undergraduate students and one faculty member on the Mexico City campus. Additionally, the teaching schedule considered two 1.5 h sessions per week during the semester. Finally, this course followed the academic regulations and policies at that moment.

2. Referring to the learning experience

The execution of the learning experience covered all elements of the ADDIE framework in Table 2, as expected. The exemplification of ADDIE intended to develop a learning experience according to the aim of this work (see Section 1).

3. Concerning the collection of observations (data) on the learning experience

- The collection of observations did not involve student gender, age, background, and attendance for this work. Referring to attendance is not an academic requirement for assessment and evaluation in the course;
- All students had the same responsibility and opportunity to participate in the learning activities, evaluations, assessments, and then, in the observation collection process, in the learning experience. This consideration means that data collection did not consider samples or a random selection of students during the execution of the learning experience;
- All collected observations (e.g., students' opinions in surveys and reports) regarding the learning experience are observation-dependent. Therefore, this learning experience is a single instance of the research object. It appears possible to elaborate concluding statements about the learning experience, but there cannot yet be generalizations in some other cases of the object;
- Students anonymously and voluntarily answered surveys, resulting in different participation rates;
- Students reported no significant disruptions to their participation in the learning experience because of the pandemic, except for one student who had limitations to engage as expected because of difficulties associated with acquiring COVID-19;
- There is no evidence in this work of students' work or learning outcomes in the learning experience, as they were not part of this work.

### 3.2.3. Check Stage

This stage relates to collected observations regarding the students' results of their evaluations, student outcome attainment, and opinions regarding the learning experience. This work does not include specific exams, report briefs, and assessment rubrics due to confidentiality. Table 5 exhibits the results of evaluations via exams and project reports for

the 16 students. Table 6 collects the results of the instructor's assessment of student learning outcomes, while Table 7 presents the results of the student opinion survey regarding the learning experience. Finally, Table 8 contains the results from the institutional student opinion survey ("Encuesta de opinión de alumnos" (ECOA) in Spanish) about the course. The ECOA summarizes descriptive statistics based on students' answers from the academic administration. Tables 5–8 present descriptive statistics to describe the data distribution, such as the means, standard deviations (Std Dev), medians, and interquartile ranges (IQR). Statistical analysis commonly begins with calculating descriptive statistics to characterize the features or attributes of the collected data in tables or graphs [142].

**Table 5.** Student learning evaluations.

| Evaluation | 1st Partial Exam | 2nd Partial Exam | Final Exam | Partial Project Report #1 | Partial Project Report #2 | Final Project | Final Score/Grade |
|---|---|---|---|---|---|---|---|
| Mean | 94.31 | 92.69 | 53.44 | 94.63 | 96.25 | 97.50 | 84.98 |
| Std Dev | 12.18 | 15.13 | 27.37 | 7.34 | 8.74 | 9.68 | 10.71 |

Evaluation scale (0–100), minimum passing mark 70, 87.5% pass rate.

**Table 6.** Assessment of student learning outcomes.

| Student Learning Outcome | ABET (H) | ABET (K) | Citizenship Commitment |
|---|---|---|---|
| Median | 3 | 2 | 3 |
| MIN | 1 | 1 | 1 |
| MAX | 3 | 3 | 3 |
| Q1 | 3 | 2 | 3 |
| Q3 | 3 | 2 | 3 |
| Interquartile Range (IQR) | 0 | 0 | 0 |
| Achievement level 2 or above | 93.75% | 87.5% | 93.75% |
| Achievement level 3 | 87.5% | 18.75% | 93.75% |

Attainment level (0 = not able to be assessed, 1 = Below acceptable, 2 = Minimum acceptable, and 3 = Exceeding acceptable.

**Table 7.** Learning experience student opinion survey.

| Learning Experience Student Opinion Survey Student Answers—Initial Survey: 12 out of 16 (75%) Student Answers—Final Survey: 16 out of 16 (100%) | Relevance | | Interest | | Motivation | | Student Learning Outcome Level of Attainment | |
|---|---|---|---|---|---|---|---|---|
| | Initial | Final | Initial | Final | Initial | Final | Initial | Final |
| Median | 5 | 5 | 5 | 5 | 5 | 5 | 4 | 5 |
| MIN | 4 | 4 | 4 | 4 | 3 | 2 | 3 | 4 |
| MAX | 5 | 5 | 5 | 5 | 5 | 5 | 5 | 5 |
| Q1 | 4 | 5 | 4 | 5 | 4 | 4 | 3 | 4.25 |
| Q3 | 5 | 5 | 5 | 5 | 5 | 5 | 5 | 5 |
| IQR | 1 | 0 | 1 | 0 | 1 | 1 | 2 | 0.75 |
| $p$-value (Mann–Whitney two-tailed test, significance level $\alpha = 0.05$) | 0.772 | | 0.149 | | 0.596 | | 0.069 | |

Evaluation Likert scale (1 = Poor and 5 = High), see Table A1 for the questions in Appendix A.

**Table 8.** Institutional student opinion survey (ECOA) results.

| Institutional Student Opinion Survey # Student Answers: 11 out of 16 (68.75%) | 1. MET | 2. PRA | 3. ASE | 4. EVA | 5. RET | 6. APR | 7. DOM | 8. REC | 9. COM (7 Student Comments) |
|---|---|---|---|---|---|---|---|---|---|
| Mean | 10.00 | 10.00 | 10.00 | 9.91 | 10.00 | 10.00 | 10.00 | 9.36 | 100% of comments highlight support, clarity of explanations, applications, commitment, and knowledge proficiency. |
| Std Dev | 0 | 0 | 0 | 0.29 | 0 | 0 | 0 | 1.49 | |

Evaluation Likert scale (0–10), the target result is 9.0 minimum. The REC mean value at the school level is 8.91.

Moreover, the results show that the students achieved an 87.5% passing rate based on their evaluation, 93.75% attained a minimum acceptable level in the ABET student outcome (H), and 87.5% in (K). Finally, 93.75% obtained a minimum acceptance level in citizenship commitment to social transformation. The ECOA survey results reveal that all values exceeded the targets (+9.0 on a 0–10 scale). The results of a study on the impact of the COVID-19 crisis on food chains corresponding to academic work are presented elsewhere [129].

The notation for Table 8 is as follows.

- MET—Teaching methodology and learning activities (0 = Very poor and 10 = Exceptional);
- PRA—Concept comprehension based on practical applications (0 = Very poor and 10 = Exceptional);
- ASE—Tutoring (0 = Very poor and 10 = Exceptional);
- EVA—Evaluation and feedback (0 = Very poor and 10 = Exceptional);
- RET—Intellectual challenge (0 = Very poor and 10 = Exceptional);
- APR—Instructor support and commitment (0 = Very poor and 10 = Exceptional);
- DOM—Knowledge proficiency (of the instructor) (0 = Definitively no and 10 = Definitively yes);
- REC—Course recommendation (0 = Definitively no and 10 = Definitively yes);
- COM—Students' comments.

The results of students' work on SCM and system dynamics explored the complications and difficulties of the pandemic shedding light on complex issues. For instance, students investigated the effects of supply interruptions on low socioeconomic levels in the population, the increased consumption of non-nutritious food, and the decrease in food quality and availability. They also explored the increase in delivery times, the shift from traditional supply formats to online and telecommunication-based alternatives, the (re-) configuration of the retail landscape, and the transformation of business and supply chain models to catch up with the changes in demand (see [129] for further details). Finally, students also developed strategies for improving the food supply, especially for strengthening the role of nanostores and local markets in neighborhoods for community food resilience, fighting the bullwhip effect in food chains to minimize inventory oscillations in retailers, and using information and communication technologies to improve food accessibility.

### 3.2.4. Act Stage

The results from the learning experience suggest adequate passing rates, student learning outcome assessments, and the institutional student opinion survey. All comments were positive and provided consistent feedback on surveys, evaluations, and assessments. The results of the learning experience survey are further discussed in the next section to further explore the implications of this work. Hence, there is no suggestion for improvement actions for further implementations of the learning experience.

## 4. Discussion

### 4.1. Results Discussion

The results from the learning experience suggest progress toward the aim of this work and answering the research question (see Section 1). The results from the student opinion surveys in Table 7 show that students regarded the learning experience as highly relevant, motivating, and interesting, with low levels of (or none) variability in their opinions. All the median values reached the top value of 5 in the final survey, and the IQR decreased from 1 to 0 in relevance and interest, whereas it remained constant at 1 for motivation. Concerning the attainment of the citizenship commitment ability, the median increased from 4 to 5, and the IQR decreased from 2 to 0.75. However, the Mann–Whitney test was used to examine these results more deeply and determine whether the two groups' population medians differed, assuming that the data had a similar shape and spread, and did not need a normal distribution [143]. The *p* values of the Mann–Whitney two-tailed test indicated that the null hypothesis could not be rejected because $p > \alpha = 0.05$. This result means that the medians of the two groups of survey answers were not different (H$_0$: $\theta_x = \theta_y$, H1: $\theta_x \neq \theta_y$, where $\theta_x$ is the median of the first group and $\theta_y$ is the median of the second group). We could interpret this as a considerable number of students recognizing that the learning experience contributed to improving their ability. This improvement might result from the substantial decrease in the IQR during the learning experience throughout the semester.

Regarding educational approaches that create engaging and participatory learning experiences, this notion might link to the MET question of the ECOA institutional survey about the teaching methodology and learning activities. The students answering this survey considered the learning activities to be exceptional (see Table 8). The average value achieved on this question was 10.00, and the standard deviation was 0.00. However, the survey had no students' comments about the learning activities or their participation and engagement.

Referring to the evaluation and assessment tools to learn about the students' views regarding the learning experience, we can use the results and descriptive statistics from the students' answers to questions PRA, ASE, EVA, RET, and APR of the ECOA institutional survey in Table 8. Except for EVA (with a mean value of 9.91 and a standard deviation of 0.29), all other results achieved a mean value of 10.00 and a standard deviation of 0.00. Additionally, the REC results suggested a mean value of 9.36 and a standard deviation of 1.49. All of these values fall above the target of 9.00 and the REC mean value of 8.96 at the school level. It is necessary to mention that no additional information is available to understand the cause of the deviation, as students answered the survey anonymously. Thus, there is a need to develop or adapt new instruments to provide specific details of the learning experience and the experiential learning activities. For instance, the institutional survey relates to courses and not learning experiences, and the types of experiential learning activities are not explicit anywhere.

Concerning the student learning evaluations, the results presented in Tables 5 and 6 indicate the outstanding marks and attainment levels in the student learning outcomes. The average marks of this course did not deviate from those obtained in the corresponding department and the School of Engineering and Sciences in Mexico City. Nevertheless, the assessment of ABET student outcome K raises concerns because just 18.75% achieved an exceeding level (3). IN2005 System Dynamics is a seventh-semester course in the last third of the academic program, which still gives way for students to improve their learning outcomes in further courses and learning experiences. The implementation of improvement can occur through the ABET student learning outcome plan already in place for the IIS program in Mexico City. This plan covers, for instance, assessment center challenges in developing and assess students' competencies according to expected learning outcomes. Additionally, the COVID-19 crisis caused difficulties for just one student, who fell short in the evaluations and assessments, influencing the overall group results.

In summary, the survey results suggest that the experiential learning activities created a satisfactory learning experience for the students. This idea relates to the experiential learning theory, which claims to support motivation, participation, and engagement [96]. From the beginning of the learning experience, students recognized the importance of the study situation in the impact of supply chains on the sustainability of cities and communities, which turned out to be critical for capturing their attention and participation in upcoming activities. It also was crucial to provide the support and help students required for the learning experience according to the APR question results and positive comments in Table 8. Therefore, the overall results presented in Tables 5, 6 and 8 suggest having a motivating, interesting, and relevant learning experience with engaging and participatory activities that produced the expected learning results in student outcome development. These results are consistent with previous work on relevance, interest, and motivation, as presented in Section 1 [20,22–25].

In the research process, collected observations and interpretations require discussions regarding *reliability*, *transferability*, and *validity*. Concerning *reliability*, it is not yet possible to claim that observations can consistently refer to other instances of the same type because a single case study was used. Therefore, future instances are required to collect additional observations, compare them, and identify similarities or invariances among observations of the learning experiences. Regarding *transferability*, this work presents a framework, a methodology, and a set of assumptions to recreate and observe new instances of the learning experiences, as presented in Section 2. Nevertheless, other researchers should develop further examples of learning experiences to exemplify the use of this work.

Finally, student evaluations and assessments had variations in their level of participation across the different surveys. The collected observations represented 68.75% of the participation in the ECOA institutional survey and 75% in the initial and 100% in the final learning experience survey. According to the Yamane simplified formula of proportions for survey answers [144], these results represent a *level of precision (e)* or *sampling error* of 16.85% for the ECOA, 14.43% for the initial, and 0% for the final learning experience survey. These calculations assume a *confidence level (P)* of 95%, *population (N)* of 16, and *sample size (n)* referring to the answers gathered in each survey. A value of 0% in precision indicates 100% accuracy in the results.

Moreover, these observations are observation-dependent on the learning experience [78,116]. Hence, these results suggest limited *validity* in claiming that observations consistently refer to the learning experience because of the variations in the survey participation level. Any interpretations of the results cannot yet be used to make inferences or claims about other learning experiences, as further instances are required.

Concerning students' work on the system dynamics applications, the results students produced aligned with the current research on the disruption of food chains during the pandemic despite data access and analysis limitations. Students created novel causal-loop models and systems archetypes to understand mental models of the study situation and identify leverage points for policymaking. Their work allowed the students to link a challenging real-world problem, their reality, the disciplinary learning activities, and the research process. Students displayed the ability to conduct research and made this effort part of an experiential learning process to benefit their future professional development. This learning experience exemplifies the possibility of undertaking *teaching-based research* with undergraduate students and expanding application cases of system dynamics, SCM, and sustainability education, as previously reported elsewhere [129].

### 4.2. Findings and Implications

This work provided a framework and exemplified its use to conceptualize learning experiences based on experiential learning in SCM education for sustainable development. The learning experience turned out to be relevant, motivating, and interesting for the students. This framework contributes to education and teaching practice in the discipline to inform *how to learn* about sustainability-related learning experiences, as little work exists

in the literature on Higher Education [32,36,52]. It also contributes to defining *what to learn* about the impact of supply chains on sustainability concerning the SDG in cities and their communities, particularly beyond technical and environmental aspects (see [50,51]). Section 2 presents an exploration of possible learning situations to guide this effort. In this direction, an application case study described a learning experience of studying food supply chains using system dynamics during the COVID-19 sanitary emergency. Nevertheless, further examples remain possible according to SCM's topics, contexts, and challenges.

Moreover, the presented learning experience provided the opportunity of going beyond the economic and environmental aspects of sustainability to explore the social dimension of supply chains. By looking at the effects of the disruption of supply chains during the pandemic, students could identify the impact on people's food supply and potential implications on their health condition and social inclusion and equity within their cities and communities.

Regarding students, this work contributed to providing a learning experience that was well-perceived and highly recommended according to the results collected in the surveys (see Tables 7 and 8). This contribution indicates the efficacy of the learning experience in providing students with an appropriate means of learning. Regarding the impact on students' learning, the results were satisfactory for marks and passing rates (see Tables 5 and 6). These results might be linked to the learning experience's effectiveness in achieving the learning outcomes. However, there is no specific statistical analysis for this purpose, which might require the correlation marks and attainment of learning outcomes with the learning experience.

Furthermore, despite not being part of the application case study and the methodology, the learning experience required a higher dedication and investment of time from the instructor. There is a need to quantify this effort to determine the efficiency of these learning experiences in using academic resources. Finally, the students' work on system dynamics applications to sustainability-related challenges for SCM is an excellent opportunity for further exploring *teaching-based research* and research-based learning for developing research skills in students.

Nevertheless, limitations exist in the implemented method in the learning experience. On the one hand, data collection in the learning experience depends on the ability and capability of students to gather data, which affects data reliability consistently. On the other hand, data analysis was limited to those methods involved in the course or those that students already knew from previous courses. This limitation made it necessary to review additional methods not covered in the syllabus, affecting the course learning plan.

Additionally, a limitation existed concerning the methodological approach of this work. Using a single case study presents conclusions about one instance of the research object, namely a learning experience. The application case study explores and exemplifies the use of the presented framework to conceptualize a learning experience about sustainability issues for SCM education. Therefore, there is no possibility and intention to validate this work's impact on students' learning or the achievement of learning outcomes, nor making inferences or generalizations about other instances.

Additionally, incorporating the study situation as part of the instructional design involved a different workload and effort from the instructor during the academic term. This requirement covered the learning experience design, planning, execution, and evaluation. In this sense, this extra required effort might discourage other academics from adopting and replicating this effort.

*4.3. Future Work*

There is a need for further implementations of learning experiences to improve the design and evaluation of the research results. In terms of designing learning experiences, future work should focus on creating new instances for data collection, analysis, and the development of statements and conclusions on the framework, its contributions, and its use. This proposition should not only include instances of the same study situation, but

also different topics and scenarios about the impact of supply chains on the sustainability of cities and their communities, for instance, urban mobility, waste reduction, and health and well-being improvement. This could also involve other courses and modules at the undergraduate and postgraduate levels across different developing countries to instrument high-impact, applied research and learning experiences to enrich SCM education. Additionally, future work is required to elucidate the learning experience's effectiveness in improving students' learning. Nevertheless, this effort will require adopting, adapting, or developing new data collection and instruments for the deeper analysis of identified variables.

Moreover, there should be new explorations into the link between supply chains and the impact on the sustainability of cities and their communities to clarify the implications beyond existing SDG indicators. For instance, the effects of logistic operations on vehicle traffic, drivers' stress, and social inclusion and equity in underserved communities are being explored. This work could help to conceptualize existing challenges and identify new relevant scenarios as study situations for novel learning experiences about sustainability for SCM education.

## 5. Conclusions

This work contributes to SCM education by including sustainability-related challenges and disciplinary topics in novel learning experiences that will improve the preparation of future professionals as problem solvers and decision-makers. This view calls for developing interesting and motivating learning experiences to enhance students' engagement and participation.

This learning experience incorporates the SDG to study the impact of supply chains on the sustainability of cities and their communities to advance SCM education and teaching practice. Furthermore, the challenging learning situations can help to expand the conceptualization of application case studies based on the economic, social, and environmental aspects of sustainability and their link to SCM practices and operations in cities. Further work should be conducted on identifying new learning challenges, their implementation in learning experiences, data collection and analysis, and clarifying the link between SCM and the SDGs for educational purposes. There is also a need for future research concerning the measurement and evaluation of this work's impact on students' learning and the achievement of their outcomes in other learning experiences.

**Author Contributions:** Conceptualization, D.E.S.-N. and C.M.-A.; methodology, D.E.S.-N., C.M.-A., L.M. and E.Z.R.-C.; validation, C.M.-A., L.M. and E.Z.R.-C.; formal analysis, D.E.S.-N.; investigation, D.E.S.-N., C.M.-A., L.M. and E.Z.R.-C.; resources, L.M.; data curation, D.E.S.-N.; writing—original draft preparation, D.E.S.-N., C.M.-A., L.M. and E.Z.R.-C.; writing—review and editing, C.M.-A., L.M. and E.Z.R.-C.; visualization, D.E.S.-N.; supervision, D.E.S.-N.; project administration, D.E.S.-N.; funding acquisition, L.M. All authors have read and agreed to the published version of the manuscript.

**Funding:** The APC was funded by the Writing Lab, Institute for the Future of Education, Tecnologico de Monterrey, Mexico.

**Institutional Review Board Statement:** Ethical review and approval were waived for this study due to the review board deeming it "Research without risk," i.e., studies using retrospective documentary research techniques and methods, as well as those that do not involve any intervention or intended modification of physiological, psychological, and social variables of study participants, among which the following are considered: questionnaires, interviews, review of clinical records, and others, in which they are not identified or sensitive aspects of their behavior are not addressed.

**Data Availability Statement:** The data presented in this study are available on request from the corresponding author (D.E.S.-N.).

**Conflicts of Interest:** The authors declare no conflict of interest. The funders had no role in the design of the study, in the collection, analyses, or interpretation of data, in the writing of the manuscript, or in the decision to publish the results.

## Appendix A

**Table A1.** Questions in the learning experience student opinion survey.

| Variable | Initial Survey Questions | Final Survey Questions |
|---|---|---|
| Relevance | How RELEVANT is undertaking Sustainable Cities and Communities learning activities in this course to your studies and professional practice? | How RELEVANT was undertaking the Sustainable Cities and Communities learning activities in this course to your studies and professional practice? |
| Interest | What level of INTEREST do you gain from undertaking the Sustainable Cities and Communities learning activities in this course to benefit your future professional practice? | What level of INTEREST did you gain from undertaking the Sustainable Cities and Communities learning activities in this course to benefit your future professional practice? |
| Motivation | What level of MOTIVATION do you gain from this course's Sustainable Cities and Communities learning activities? | What level of MOTIVATION did you gain from conducting the Sustainable Cities and Communities learning activities in this course? |
| Citizenship commitment to learning outcome | How do you now consider the level of development of your ability to create committed, sustainable, and supportive solutions to social problems and needs through strategies that strengthen democracy and the common good? | 4. How do you consider the development of your ability to create committed, sustainable, and supportive solutions to social problems and needs through strategies that strengthen democracy and the common good in the Sustainable Cities and Communities learning activities in this course? |

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
