# Peer review of "Experiential Learning for Sustainability in Supply Chain Management Education"

_sustainability, doi:10.3390/su142013133_

Round 1
Reviewer 1 Report
This paper tackles interesting and relevant topic. It has very good theoretical background and sound research design. Conclusions, implications, and limitations are supported with reported findings. Considering all the aforementioned, I would argue for the acceptance of this paper.
Reviewer 2 Report
The proposal relates a learning experience in a context of sustainable development that takes up a lot of space and whose interest is not seen in the focus of the article. The experiment should work whatever the subject. Or, we do not understand the link between the subject treated and the way of treating it. There is really a lack of elements to really understand what the learning activities were. Nothing mentions the difficulties encountered or potential problems with the method used. Finally, it is an account of a well-supervised teaching, but which does not really differ from another teaching which would also have been well supervised. In fact, this work lacks a real research problematization, which is a good article presenting a teaching, but of which we do not see what it can bring in terms of research. With a consequent pruning on sustainable development which takes up a lot of space, this could be an article presenting an original teaching, provided that we have more precise examples and explanations. As it is, it's pretty general.
Reviewer 3 Report
Review article: Experiential Learning for Sustainability in Supply Chain Management Education
By: David Ernesto Salinas-Navarro1*, Christopher Mejia-Argueta2 , Luis Montesinos3, 4*, and Ericka Z. Rodriguez Calvo
In Abstract; change the style from: We exemplify the use of to: in the article…..
To much words in key words: Keywords: experiential learning; supply chain management; logistics; sustainable development goals; sustainable cities and communities; food chains; engineering education; management education; educational innovation; higher education – leave max 4/5
Abbreviation: Higher Education (HE) – it would be better to use the whole sentence: higher education
Rewrite: 60 percent to 60%.
After Tables should be added sources.
Page 13: 4. Learning outcomes – move to next page
Round 2
Reviewer 2 Report
Hello, I am having a hard time with this text. Finally, I could make exactly the same comments as for the first version. In fact, there are two topics in the article. The first concerns the relevance of teaching sustainable development from the supply chain. It is a perfectly legitimate, argued and illustrated contribution. It may lack a distancing with at least a Swot analysis of intentions. The second topic concerns experiential learning. And that is the problem. We do not really understand what it has experiential, because the activities are described very succinctly. Did the students collect data with field surveys? Did the students use and rework data from major surveys by working on analysis matrices? Did the students conduct expert interviews? Have they implemented specific data processing that involves specific learning? In short, we don't really know what they did that was particularly active in learning. We must believe that it was learning by experience. I say nothing about the number of 16 students, without a control group, because on this register we could not say really positive things. From the point of view of research, there is not really any theoretical support in this register, not even a simple evocation of old Bloom's taxonomy to characterize at least the learning activities and the specificity of the subject by relation to this backing. If it is a question of working in research on this question, I think that there is a very big job to be done to characterize exactly what this learning by experience is, to study it, and to measure the impacts. But I don't see how to do it afterwards. Or, to evoke much more succinctly this question of learning by experience, to concentrate on sustainable development and the teaching situation. But at least, by seeking to identify learning objectives more clearly and establish their place in the curriculum. However, does this fit into the perspective of this journal issue?
Round 3
Reviewer 2 Report
Hello, The authors have made efforts. I find that the conclusion uses far too many references and drowns out its project. The development is also far too referenced. With a litany of arguments from authorities that we must take like that without discussion. The conclusion is much too short and we finally learn nothing except that we must continue... I recommend a better structured introduction with clearer objectives, the results of which will be at the end of the articles. Make a slimming cure in the references. Rework the end which makes the connection between the original intentions, what we have achieved. Finally a minimum of Swot analysis, to identify anyway what works and what does not work. I've never seen anything that was completely successful without risk.